# Photoinduced Mn catalysis for efficient platform for C-heteroatom bond coupling of aryl halides

Geyang Song[1,3], Jiameng Song[1,3], Qi Li[1], Xiaoli Shi[1], Xinyi Liu[1], Deng Pan[2], Tengfei Kang[1], Jianyang Dong[1], Gang Li[1], Huaming Sun ®[1], Juan Fan[1], Chao Wang ®[1] & Dong Xue ®[1] ✉

Photoinduced transition metal catalysis offers innovative strategies for fostering novel chemical reactions and improving established ones. In this work, we present a highly efficient, photoinduced Mn(II)-bipyridine catalyzed C–N, C-O and C-S coupling reaction between aryl halides—particularly less reactive aryl chlorides—and nucleophiles containing nitrogen, oxygen, and sulfur. This protocol does not need an external photocatalyst, as the single Mn(II)–bipyridine complex simultaneously serves as both the light-harvester and the metal catalyst. This method exhibits excellent substrate scope, covering eight different nitrogen sources for C-N coupling, as well as C-O coupling with alcohols, C-S coupling with thiophenols, encompassing more than 150 examples, with yields reaching up to 94%. Mechanistic studies suggest that this reaction may be initiated and sustained by the Mn(I) species through the photoinduced homolysis of the catalyst precursor bipyridine-Mn(II)(OAc)$_2$, likely proceeding via a Mn(I)/Mn(III) catalytic cycle.

Transition metal-catalyzed C–heteroatom bonds cross-coupling of readily available aryl halides with nucleophiles are the cornerstone of modern synthesis, significantly accelerating the process of molecular diversification[1-4]. For decades, palladium catalysts have long dominated this field[5-8], but they are expensive and scarce. Therefore, earth-abundant metals provide an alternative to Pd-catalysts and have contributed to advances in sustainable chemical manufacturing[9]. In this context, nickel[10-13], copper[14-16], and cobalt[17-22]-catalyzed C–heteroatom bond coupling provides a viable and environmentally friendly approach for the development of coupling reactions (Fig. 1A).

Manganese is ranking third in Earth's crustal abundance (~950 ppm), following only iron and titanium (Fig. 1A). As an essential element for life (participating in various metalloproteins, with a daily human requirement of 2–5 mg)[23], manganese plays pivotal roles in development, metabolism, and antioxidant systems in humans[24]. Its low toxicity and eco-friendly attributes[25] brings significant advantages for purification procedures in late-stage drug synthesis[26]. In 2009, Teo

group reported the first Mn-catalyzed C-N cross-coupling reaction with aryl iodides at 130 °C[27-31]. However, subsequent studies by the Madsen group demonstrated that the observed catalytic activity was attributable to copper species present in the Mn catalyst.[32] Since then, Mn-catalyzed C–heteroatom bond coupling reactions have shown minimal advancement under thermal conditions (Fig. 1B)[27-39]. Although Mn catalysts have demonstrated notable success in various fields[40], including oxidation[41], reduction reactions[42-44], and C–H bond activation[45,46], no Mn-based catalytic method suitable for C–heteroatom coupling reactions has been developed to date. It should be noted that, compared to other late 3 d abundant metals such (Pd, Ni, Co, Cu)[47], manganese's catalytic potential in coupling reactions has not yet been fully exploited. Therefore, there is an urgent need to establish efficient Mn-based catalytic systems to fully leverage its unique advantages.

The photoinduced transition-metal catalysis has recently opened up a new paradigm for coupling reaction[10,48-51]. In this process,

[1]Key Laboratory of Applied Surface and Colloid Chemistry, Ministry of Education, and School of Chemistry and Chemical Engineering, Shaanxi Normal University, Xi'an, China. [2]School of Chemistry and Materials Science, Hangzhou Institute for Advanced Study, University of Chinese Academy of Sciences, Hangzhou, China. [3]These authors contributed equally: Geyang Song, Jiameng Song. ✉e-mail: xuedong_welcome@snnu.edu.cn

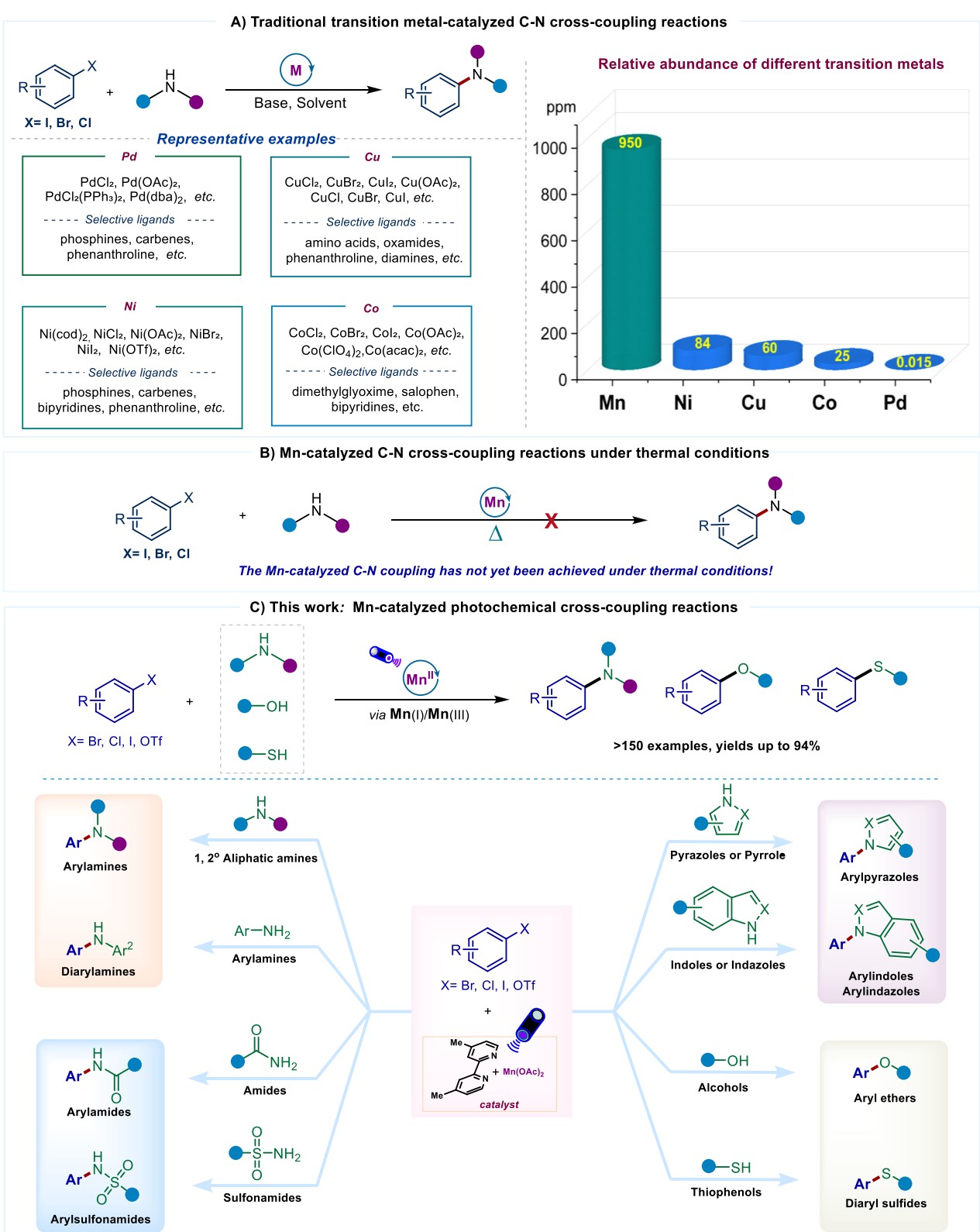

**Fig. 1 | Development of C−heteroatom bonds cross-couplings. A** Traditional transition metal-catalyzed C-N cross-coupling reaction. **B** Mn-catalyzed C-N cross-coupling reactions under thermal conditions. **C** Mn-catalyzed photochemical cross-coupling reactions.

transition metal complexes not only absorb photon energy but also facilitate the formation and cleavage of chemical bonds through a single catalytic cycle, thereby obviating the requirement for exogenous photosensitizers typically employed in conventional dual photocatalysis systems[52–56]. This novel approach has not only spawned unprecedented transformation reactions but has also improved known ones. Unlike the application of Mn-based catalysts in C−heteroatom bond coupling reactions under thermal conditions, the use of Mn complexes in photochemical reactions has attracted considerable interest among synthetic chemists[45,46,57–63]. For example,

Fadeyi group[49] first demonstrated the application of $Mn_2(CO)_{10}$ in the photochemical C-H alkylation. Subsequently, the Nagib's group[62] harnessed this complex to facilitate the coupling of ketyl radicals and carbonyl compounds. Ackermann group[46] introduced $CpMn(CO)_3$ as a photocatalyst to achieve C-H arylation of arenes in a continuous-flow photoreactor. Thomas group[58] employed the $(dmpe)_2MnBr_2$ complex for photochemical constrcont C−B bond. Recently, Xie group[59] developed $Mn_2(CO)_8Br_2$ catalyzed enantioselective $C(sp^2)$−$C(sp^3)$ bond-forming for the synthesis of skipped dienes. However, the application of Mn complexes in photochemical C−heteroatom bond cross-coupling reactions have not yet been explored. In this work, a highly efficient photocatalytic platform is established by employing a photoinduced transition metal catalysis strategy, utilizing a readily available bipyridine-Mn(II) catalyst system, thereby successfully enabling various C−N, C−O, and C−S cross-coupling reactions of aryl halides (Fig. 1C). This method features excellent reaction efficiency, extensive substrate scope, and good functional-group compatibility.

## Results and discussion

### Investigation of the reaction conditions

In our preliminary study, we used the Mn-catalyzed C-N coupling reaction of bromobenzene (1) and n-butylamine (2) as a model reaction to optimize the conditions. As shown in Table 1, screening of various Mn catalysts (Supplementary Table S1) revealed that the Mn(OAc)$_2$-4,4′-dimethyl-2,2′-bipyridyl (d-Mebpy) complex could efficiently catalyze the C-N coupling upon irradiation with a purple LED under argon, affording the target product 3 in 90% isolated yield (entry 1)[64]. Further optimization showed that the proper wavelength of light was crucial. The use of 390–395 nm light resulted in the highest yield, while other wavelengths of light gave trace amounts or no desired product (entries 2–4; Supplementary Table S3). The solvent had an obvious effect on the yield, with dimethylacetamide (DMAC) being optimal (Supplementary Table S4). The choice of base also had a substantial impact (Supplementary Table S5). Organic bases performed better than inorganic bases, with 1,8-diazabicyclo (5.4.0) undec-7-ene (DBU) having the strongest promotional effect (Supplementary Table S5, 6), and no reaction occurred in its absence (entry 5). It is worth noting that the

ligand played a key role in the reaction (Supplementary Table S7); specifically, no reaction occurred in the absence of a bipyridyl ligand (entry 6). Notably, only a trace amount of the C−N coupling product was obtained when the reaction was carried out at room temperature (entry 7). Furthermore, the reaction did not occur in the absence of light, even at high temperature (entry 8). Control experiments showed that the reaction failed to proceed in the absence of a Mn catalyst or under air (entries 5 – 10). In addition, Madsen found that trace amounts of Cu species could potentially catalyze this C−N coupling reactions[29]. To this end, we analyzed the manganese catalyst using inductively coupled plasma mass spectrometry (ICP-MS) to confirm whether any of these metals were present. The analysis results showed that Pd and Cu were indeed present at parts per billion (ppb) levels (Supplementary Table S8). Therefore, we conducted C−N coupling reactions in the presence of Pd or Cu catalysts under otherwise standard conditions (Supplementary Tables S9−S13), and found that the coupling reactions did not occur (entries 11−12)[64]. These results demonstrate that these metal contaminants did not act as catalysts in the Mn-catalyzed amination reactions. Notably, this photoinduced Mn-catalyzed C-N coupling possible proceed via a distinct mechanism pathway compared to conventional Mn-catalyzed C-N, C-O/S coupling under thermal reactions (vide infra)[33–36].

### Substrate scope of C−heteroatom couplings

With the optimized conditions in hand, we next explored the scope of the aryl halides. As shown in Fig. 2, aryl halides with various functional groups reacted with n-butylamine efficiently, delivering the desired arylamines in high yields. It should be noteworthy that adding 2.0 equiv. tetrabutylammonium iodide (TBAI) as an additive exhibits a clear promoting effect for the coupling of aryl chlorides. It may induce the conversion of aryl chlorides through halogen exchange into more reactive aryl iodides to yield C−N coupling products in relatively considerable yields (Supplementary Table S17 and S18). Specifically, para-substituented aryl halides with an electron-neutral (3), electron-donating (4–13), or electron-withdrawing (14–21) groups were all reactive, affording the desired products in good yields. It is worth highlighting that all the unactivated aryl chlorides that contain

## Table 1 | Optimization of reaction conditions[a]

| Entry | Variation from standard conditions | Yield (%)[b] |
|---|---|---|
| 1 | Standard conditions | 95, 90[b] |
| 2 | Standard conditions, UV (360–365 nm) | 25 |
| 3 | Standard conditions, blue LEDs (460–465 nm) | N.R. |
| 4 | Standard conditions, white LEDs (6500 K) | N.R. |
| 5 | Standard conditions, no DBU | N.R. |
| 6 | Standard conditions, no ligand | N.R. |
| 7 | Standard conditions, light, r.t. | trace |
| 8 | Standard conditions, no light, r.t. or 120 °C | N.R. |
| 9 | Standard conditions, no Mn catalyst | N.R. |
| 10 | Standard conditions, air, light | N.R. |
| 11 | Standard conditions, Cu salt | N.R. |
| 12 | Standard conditions, Pd salt | N.R. |

[a]Standard conditions:Bromobenzene (1, 0.2 mmol), n-butylamine (2, 2.0 equiv., 0.4 mmol), Mn(OAc)$_2$ (10.0 mol%), d-Mebpy (10.0 mol%), (DBU, 1.5 equiv., 0.3 mmol), DMAc (2.0 mL), purple LEDs (390–395 nm), 85 °C, under Ar, 24 h. Yields were determined by $^1$H NMR spectroscopy with 1,3-benzodioxole as an internal standard. N.R. = no reaction, r.t. = room temperature. [b]Isolated yield. For details, see Supplementary Tables S1–S16.

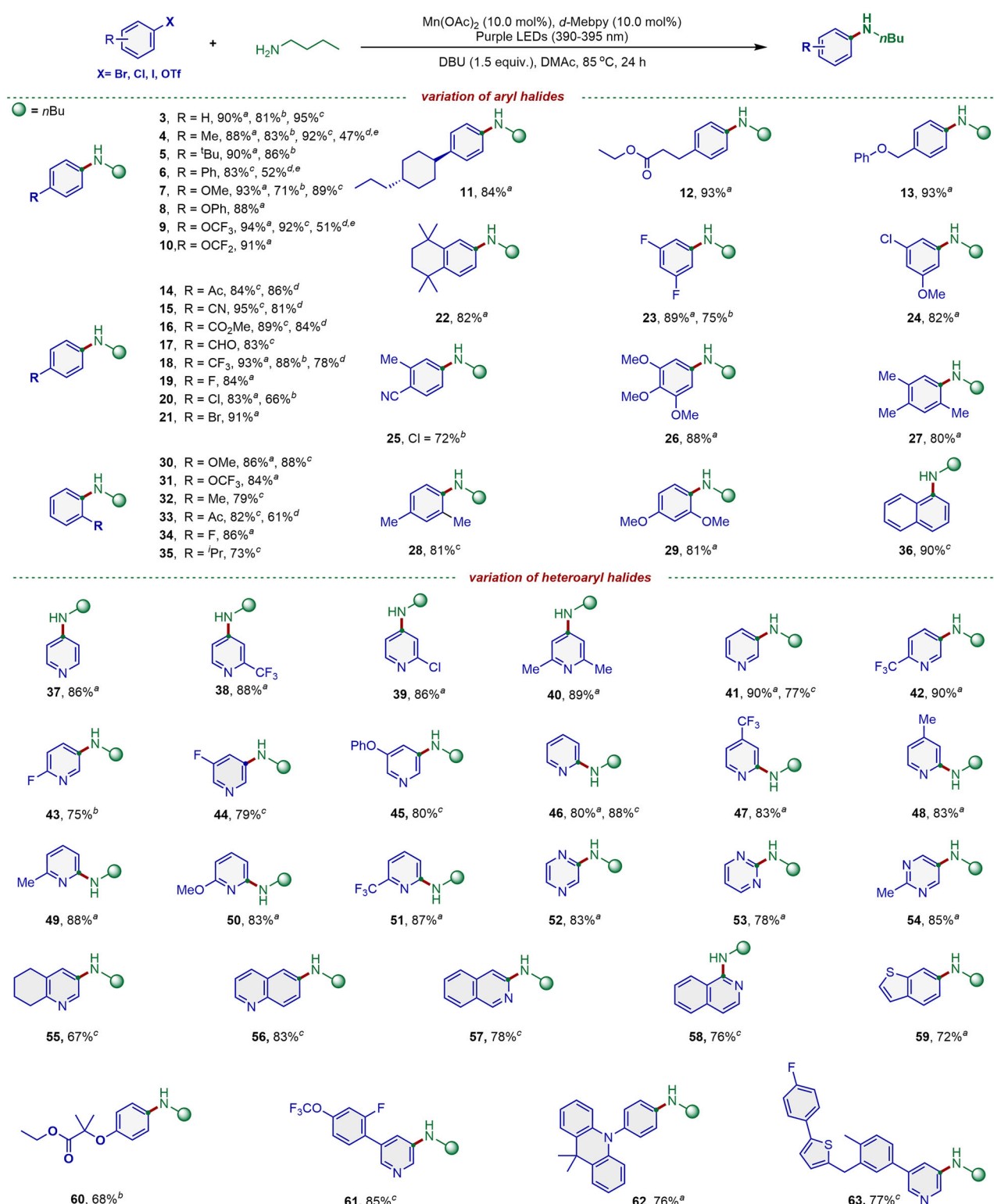

**Fig. 2 | Scope of aryl halides.** Reaction conditions: [a]aryl bromide (0.2 mmol), *n*-butylamine (2.0 equiv., 0.4 mmol), Mn(OAc)$_2$ (10.0 mol%), *d*-Mebpy (10.0 mol%), DBU (1.5 equiv., 0.3 mmol), DMAc (2.0 mL), purple LEDs (390–395 nm), 85 °C under Ar, 24 h. [b]Aryl chloride (0.2 mmol), *n*-butylamine (2.0 equiv., 0.4 mmol), Mn(OAc)$_2$ (10.0 mol%), *d*-Mebpy (10.0 mol%), DBU (1.5 equiv., 0.3 mmol), TBAI (2.0 equiv.). [c]Aryl iodides (0.2 mmol), *n*-butylamine (0.4 mmol), Mn(OAc)$_2$ (5.0 mol%), *d*-Mebpy (5.0 mol %), DBU (1.5 equiv., 0.3 mmol), DMAc (2.0 mL), purple LEDs (390–395 nm), 85 °C, under Ar, 24 h. [d]Aryl OTf (0.2 mmol), *n*-butylamine (0.4 mmol), Mn(OAc)$_2$ (15.0 mol%), *d*-Mebpy (15.0 mol%), DBU (1.5 equiv., 0.3 mmol), DMAc (2.0 mL), purple LEDs (390–395 nm), 85 °C, under Ar, 24 h. [e]48 h. Isolated yields are reported. For details, see Supplementary Information.

electron-donating substituents, such as -Me, -*t*Bu, -OMe and -OPh, are compatible in this protocol. Aryl chlorides with electron-withdrawing substituents, such as -CF$_3$, -COMe, -CN, -CO$_2$Me, and halogens in the para position of the phenyl ring, delivered the desired products with excellent yields. Noteworthy, aryl halides bearing two halogens afforded a single C–N coupling reaction (**19–21**). Polysubstituted aryl halides were also suitable, giving the corresponding products (**22–29**) in good yields. Pleasingly, *ortho*-substituted substrates afforded the desired products (**27–35**) in 61–88% yields, indicating thatsteric hindrance had minimal effect in our catalytic system. For fused-ring naphthalene (**36**), a high yield of the coupling product can also be obtained. Because heteroaromatic structural motifs are widely present in many pharmaceutical compounds, we were pleased to find that bromopyridines (**37–51**), bromopyridazine (**52**), bromopyrimidines (**53–54**), quinoline (**55–56**), isoquinoline (**57–58**), bromobenzothiophene (**59**), and bromoacridine (**60–63**) were also amenable under the coupling conditions. To further broaden the application of aryl halides, highly reactive aryl iodides also furnish C–N coupling products in high yields. Likewise, aryl sulfonates, as aryl halide analogs, show good substrate compatibility and provide the desired coupling products in moderate to good yields. These results indicate that the present Mn-catalyzed system offers broad electrophile compatibility, tolerating both halides and pseudohalides.

Next, we examined the scope of the *N*-containing nucleophiles. As shown in Fig. 3, various amines were tolerated, providing the desired *N*-arylation products in excellent yields upon reaction with 1-bromo-(4-*tert*-butyl)-benzene. Specifically, primary amines with alkyl (**64–67, 80–82**), alkenyl (**68–69**), alcohol (**70**), ether (**71, 72, 74, 83**), acetals (**73–77**), ester (**78**), cyano (**79**), furan (**84**), fluorine atom (**85**), trifluoromethyl (**86–87**), methyl sulfonyl (**88**) groups and branched alkyl chains (**89–97**) were also suitable, affording the corresponding coupling products in good yields. It is worthy of note that no C–O coupling products were observed under the standard conditions when the substrate contained both an exposed NH$_2$ group and a OH group (**70** and **97**). Moreover, the primary amine preferentially underwent the coupling reaction to give a monoaminated product (**90**) when a secondary amine coexisted. Secondary amines also underwent the reaction, giving the desired products (**98–104**) in good yields. Substituted aromatic amines were also suitable substrates, delivering the desired diarylamine products (**105–110**) in high yields. Pleasingly, amides were also amenable to undergo the C–N coupling, thus giving the amination products (**111–119**) in high yields. It is noteworthy that upon fine-tuning of the Mn catalyst also demonstrated excellent compatibility with sulfonamide compounds, including a range of different aliphatic sulfonamides and aromatic sulfonamides bearing either electron-donating or electron-withdrawing substituents, all of which afforded the corresponding products (**120–141**) in 65–82% yields. Notably, nitrogen-containing heterocycles often pose challenges in transition-metal-catalyzed reactions due to their strong coordination ability and the resulting deactivation of the catalyst[65]. But in our protocol, heterocycles such as pyrazole (**142–146**), indazole (**147**), pyrrole (**148**), and indoles (**149–150**) produced the desired products in 53–91% yields.

To further expand the generality of this photoinduced Mn-catalyzed cross-coupling reaction, we also investigated cross-coupling reactions involving oxygen- and sulfur-containing nucleophiles (Fig. 4). Firstly, we investigated the C–O coupling of 4-bromobenzonitrile with alcohols, and the desired aryl ethers (**151–156**) were successfully obtained. Subsequently, thiophenols proved to be viable nucleophiles for C–S cross-coupling (**157–165**). Notably, even sterically hindered 2,6-substituted benzenethiols (**165**) exhibited excellent compatibility with this method. To comprehensively evaluate the general applicability of our approach, we extended the reaction scope to include a diverse range of bioactive

molecules containing C(sp$^2$)–Br moieties (**166–171**). We were delighted to find that amination products derived from pharmaceutical compounds—including estrone (**166**), gemfibrozil methyl ester (**167**), tocopheryl ether (**168**) and celecoxib with various *N*-nucleophiles (**169–173**) could be accessed in 69–88% yields. The results described above demonstrate again that the Mn-catalyzed photochemical C–heteroatom coupling has good substrate suitability, showing potential utility in synthetic and medicinal chemistry.

## Mechanistic investigation

We next conducted the control experiments aiming to understand the reaction mechanism (Fig. 5). First, we investigated the possible Mn catalyst formed between bipyridine and Mn(OAc)$_2$. We obtained a trinuclear Mn complex **174**, and fortunately, its structure was determined by single-crystal X-ray diffraction analysis (Fig. 5A). The UV–vis absorption spectrum of the Mn complex **174** showed markedly different characteristics in the long-wave UV-vis absorption region of 300–600 nm (Fig. 5A, right). In addition, the complex **174** showed an absorption peak near 386 nm that increased in intensity with increasing concentration (Supplementary Figs. S4, 5), and this complex was probably the light-absorbing species[64]. When complex **174** was used as a catalyst to catalyze the C–N coupling of 3,5-dimethylbenzene and n-butylamine, the desired product was obtained in 89% NMR yield under standard conditions, suggesting that the complex was possible the catalyst precursor in the reaction (Figs. 5B, 1). Furthermore, when complex **174** was irradiated with 390–395 nm purple light with *N*-tert-butyl-α-phenylnitrone (PBN) as a radical trap (Figs. 5B, 2), the formation of the spin adduct of the radical signal was observed by electron paramagnetic resonance (EPR) spectroscopy, exhibiting a signal with g value = 2.001; (Supplementary Fig. S7)[64]. This EPR signal may be a spin adduct of the OAc radical, with its characteristic hyperfine coupling constants consistent with the literature data[66,67]. The formation of the OAc radical indicates that the Mn−O bond may undergo photoinduced homolytic cleavage, and may simultaneously generate Mn(I) species[68,69]. Given the complexity and diversity of the photocatalytic reaction process, this Mn complex **174** nonetheless provides valuable insights into understanding the reaction mechanism. Next, we explored the possibility of Mn(I) species serving as the initiating species in the reaction. The complex **174** was irradiated with purple light for 2 h in the absence of the aryl halides and amine, which was expected to generate a Mn(I) complex. Then the aryl halides and amine then were introduced, and the reaction was allowed to proceed for 24 h in the dark, affording the corresponding aryl amine with 29% NMR yield (Fig. 5C, left). Furthermore, when a commercially available Mn(I) complex CpMn(CO)$_3$ was used as the catalyst, the target C−N coupling product was obtained in 56% yield with a bipyridine ligand under thermal conditions (in the absence of light) (Fig. 5C, right). Finally, we examined the involvement of Mn(I) species in the oxidative addition and reductive elimination processes (Fig. 5D). Inspired by the work of Khusnutdinova group[70] on the C−C bond elimination reactions in Mn(III)-aryl complexes, the Mn(III) complex **176** was synthesized by the reaction of aryl bromide **175** with Mn(CO)$_5$Br at room temperature (in a dimethylformamide solution, without the need for additives). Subsequently, an equivalent amount of *n*-butylamine was added, hope to obtain a Mn−amine complex **177**. Unfortunately, attempts to isolate or identify the metal-amine intermediate were unsuccessful, as the intermediate may rapidly undergoes reduction elimination, resulting in the expected C−N coupling product **178**. The desired C−N coupling product **178** was characterized by crude NMR and HRMS. These observations support the view that this C-N cross-coupling possibly proceeds via a Mn(I)/Mn(III) cycle, which involves the oxidative addition of aryl halides to a Mn(I) species, and the reductive elimination of an aryl-Mn(III)-amido species, thereby forming a C-N bond. Furthermore, the photo-switching experiment (Supplementary Fig. S8)

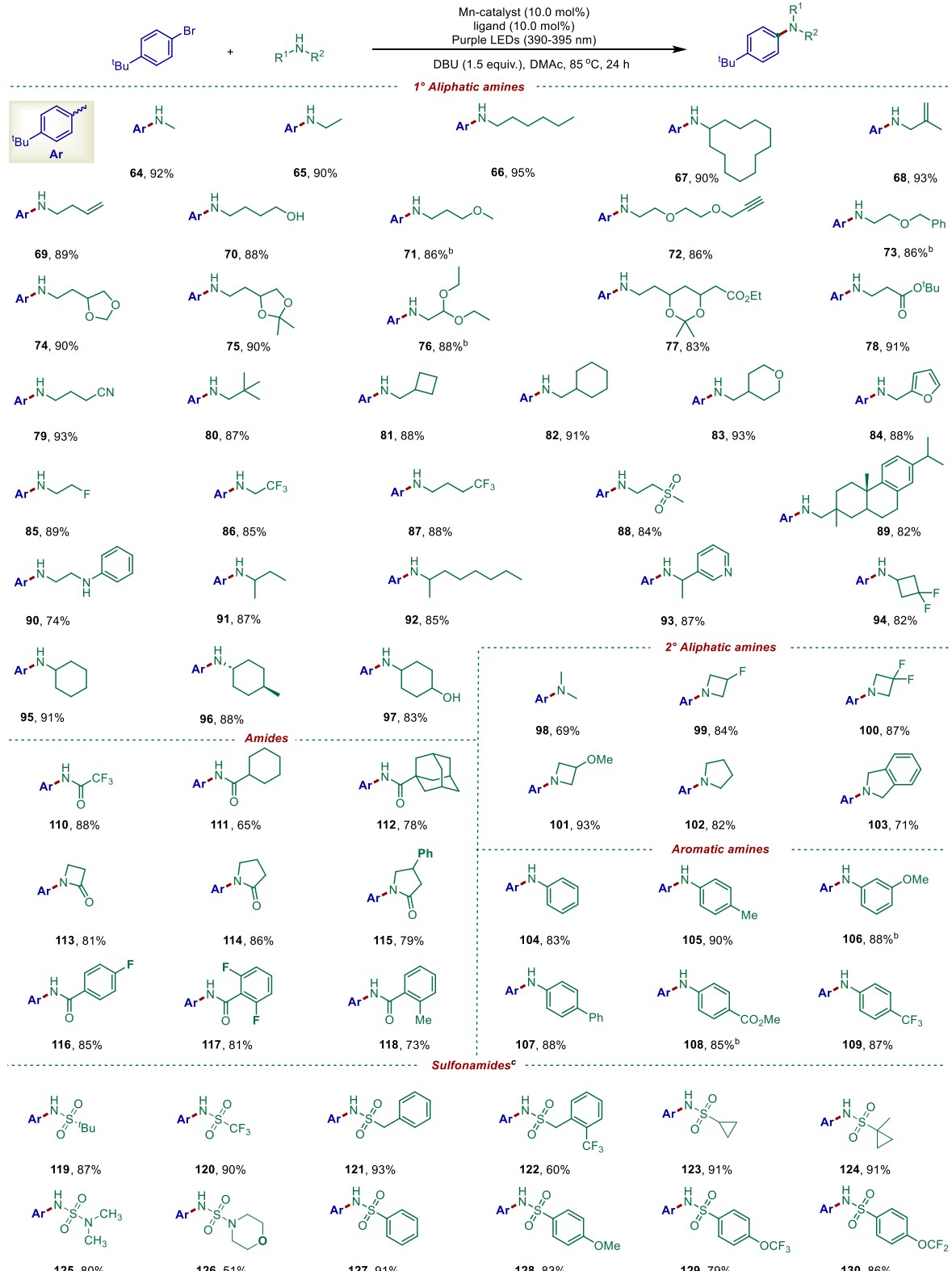

**Fig. 3 | Scope of *N*-nucleophiles for aliphatic amines, amides, arylamines, sulfonamides.** Reaction conditions: [a]aryl halide (0.2 mmol), amine (2.0 equiv., 0.4 mmol), Mn(OAc)$_2$ (10.0 mol%), *d*-Mebpy (10.0 mol %), DBU (1.5 equiv., 0.3 mmol), DMAc (2.0 mL), purple LEDs (390–395 nm), 85 °C, under Ar, 24 h.

[b]Reaction time, 36 h. [c]Mn(acac)$_2$ (10.0 mol %), *d*-OMebpy (10 mol %), [t]BuTMG (4.0 eq), DMF: [t]BuOH (1:3), purple LEDs (390–395 nm), 85 °C under Ar, 24 h. [d]DMF: PhMe (1:1). [e]Isolated yield. For details, see Supplementary Information.

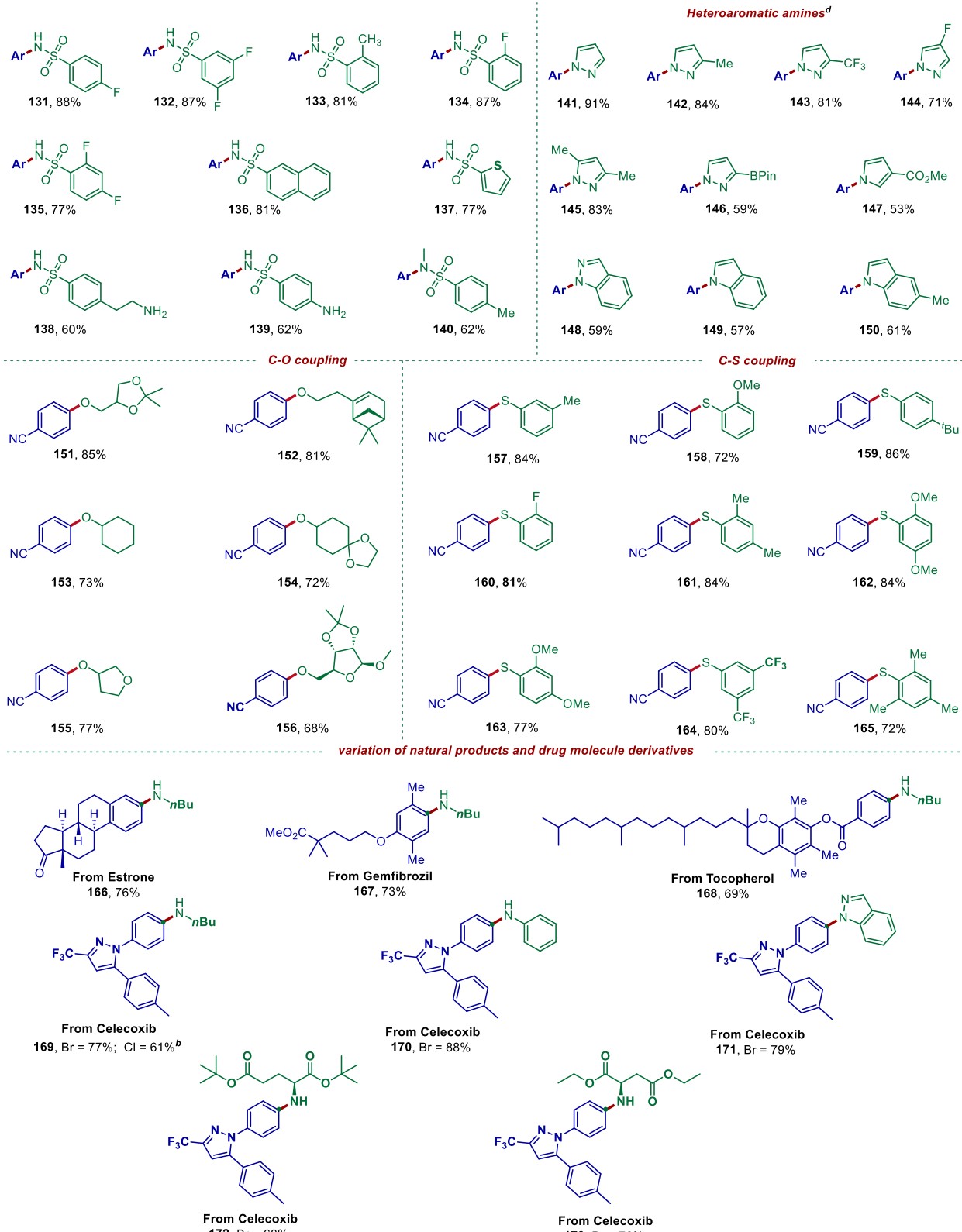

**Fig. 4 | Scope of nucleophiles for sulfonamides, heteroaromatic amines, alcohols, thiophenols.** Reaction conditions: [a]aryl halide (0.2 mmol), amine (2.0 equiv., 0.4 mmol), Mn(OAc)$_2$ (10.0 mol%), *d*-Mebpy (10.0 mol %), DBU (1.5 equiv., 0.3 mmol), DMAc (2.0 mL), purple LEDs (390–395 nm), 85 °C, under Ar, 24 h. [b]Reaction time, 36 h. [c]Mn(acac)$_2$ (10.0 mol %), *d*-OMebpy (10 mol %), [t]BuTMG (4.0 eq), DMF: [t]BuOH (1:3), purple LEDs (390–395 nm), 85 °C under Ar, 24 h. [d]DMF: PhMe (1:1). [e]Isolated yield. For details, see Supplementary Information.

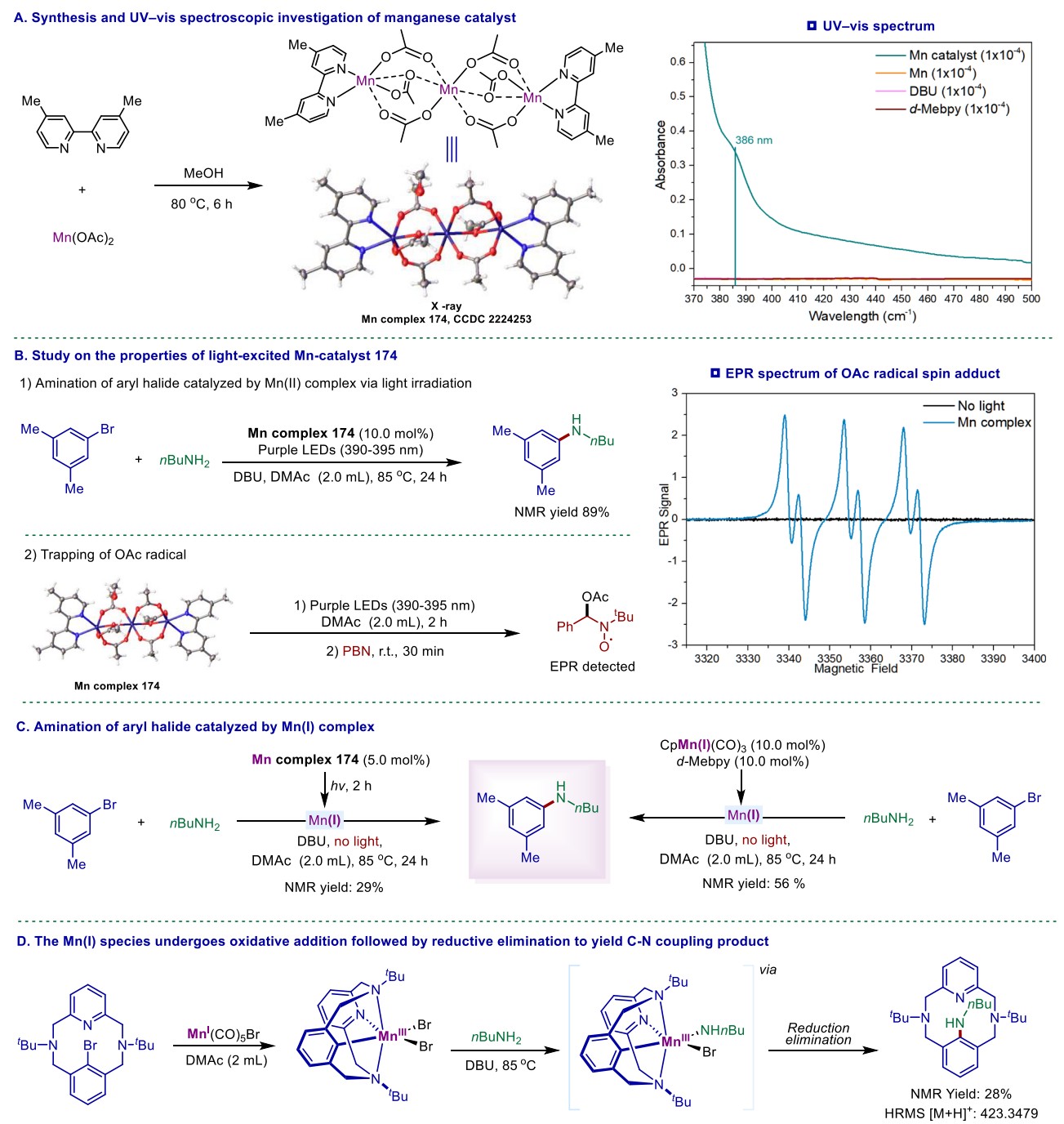

**Fig. 5 | Mechanistic investigation. A** Synthesis and UV–vis spectroscopic investigation of Mn catalyst. **B** Study on the properties of light-excited Mn-catalyst 160 and EPR spectrum of spin adduct (g = 2.001, aN = 14.56 G, aH = 2.79). **C** Amination of aryl halide catalyzed by Mn(I) complex, (**D**) The Mn(I) species undergoes oxidative addition reductive elimination of the Mn(III) species. The yields shown in the panel were determined by [1]H NMR yield using 1,3-benzodioxole as an internal standard. For details, see the Supplementary Information.

demonstrated that continuous irradiation is required to maintain the catalytic cycle, possibly to generate active Mn(I) catalysts from an off-cycle and inactive Mn(II) species.

Based on the mechanistic studies described above and the recent related reports[71–77], we propose the possible reaction mechanism shown in Fig. 6. Firstly, photoexcitation of bipyridine-Mn complex generates a Mn(I) species **I** through Mn–O bond homolysis, which undergoes an oxidative addition with the aryl halide to form a Mn(III)–Ar intermediate **II**, followed by coordination of amine to give intermediate **III**. This intermediate **III** undergoes facile reductive

elimination to afford the cross-coupling product and regenerate the Mn(I) species for the next catalytic cycle.

In conclusion, we have developed a highly efficient, photoinduced Mn(II)-bipyridine catalyzed cross-coupling reaction between aryl halides with nucleophiles containing nitrogen, oxygen, and sulfur. This protocol employs a single Mn catalyst to realize the dual roles of light harvesting and organometallic catalysis, offering an excellent substrate scope, giving synthetically and pharmaceutically useful coupling adducts in good to excellent yields. Preliminary mechanistic studies suggest that this reaction may be initiated and sustained by the Mn(I)

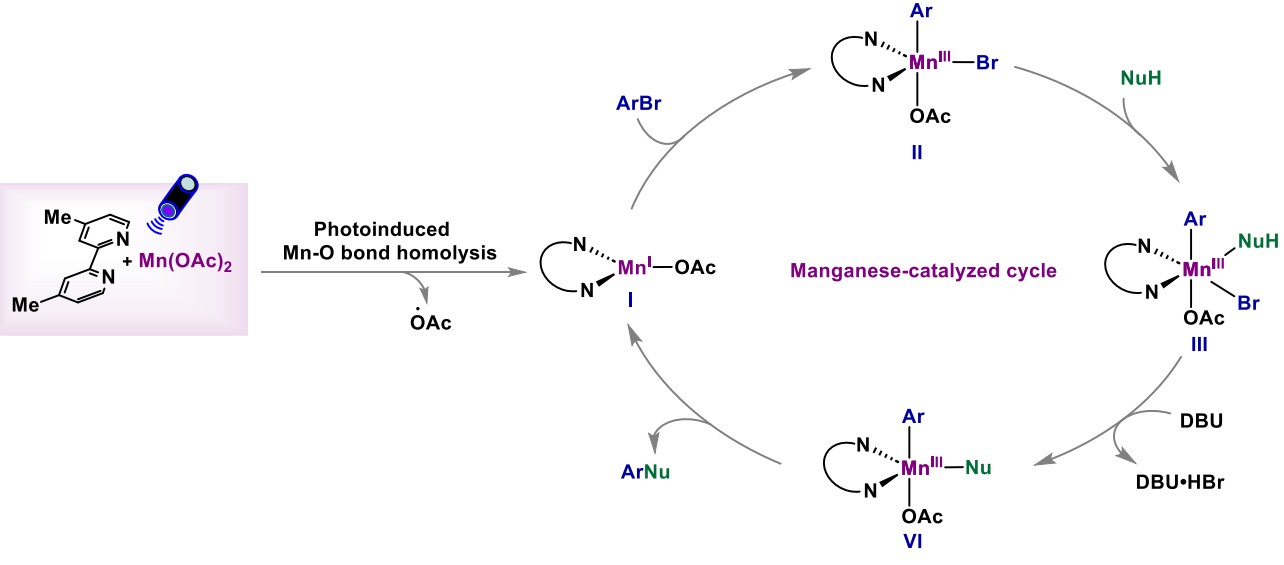

**Fig. 6 | Proposed mechanism.** The catalytic cycle for the cross-coupling under light-promoted Mn catalysis.

species, through the photoinduced homolysis of the catalyst precursor bipyridine-Mn(II)(OAc)$_2$, likely proceeding via a Mn(I)/Mn(III) catalytic cycle. The study not only improved known reactions but also opened new avenues for the application of Mn catalysis in synthetic chemistry.

## Methods

### Standard procedure for C-heteroatom bond cross-coupling of aryl halides

To an oven-dried 10 mL of storage tube were added Mn(OAc)$_2$ (10.0 mol%), $d$-Mebpy (4,4′-dimethyl-2,2′-bipyridine) (10.0 mol%), and 2 mL of DMAc with a magnetic stir bar under argon atmosphere. The mixture was evacuated and backfilled with Argon for 3 times. Then the aryl halides (0.2 mmol), $n$-butylamine (0.4 mmol) and DBU (1.5 equiv., 0.3 mmol) or TBAI (2.0 eq.) were added. The tube was sealed with the Teflon screw valve. The reaction mixture was then irradiated with 9 W purple LEDs (0.5 cm away from the tube, optical power: 320–340 mW/cm$^2$) at 85 °C. After the reaction was completed, the mixture was diluted with ethyl acetate and cooled to room temperature. The organic phases were washed with saturated ammonium chloride (3 × 10 mL), dried over anhydrous sodium sulfate, and concentrated under reduced pressure. The residue was purified by flash column chromatography using petroleum ether and ethyl acetate as eluent to afford coupling products.

### Standard procedure for heteroatom bond cross-coupling of nucleophiles

To an oven-dried 10 mL of storage tube were added Mn(OAc)$_2$ (10.0 mol%), $d$-Mebpy (4,4′-dimethyl-2,2′-bipyridine) (10.0 mol%), and 2 mL of DMAc with a magnetic stir bar under argon atmosphere. The mixture was evacuated and backfilled with Argon for 3 times. Then the aryl halides (0.2 mmol), NuH (0.4 mmol), and DBU (1.5 equiv., 0.3 mmol) were added. The tube was sealed with the Teflon screw valve. The reaction mixture was then irradiated with 9 W purple LEDs (0.5 cm away from the tube, optical power: 320–340 mW/cm$^2$) at 85 °C. After the reaction was completed, the mixture was diluted with ethyl acetate and cooled to room temperature. The organic phases were washed with saturated ammonium chloride (3 × 10 mL), dried over anhydrous sodium sulfate, and concentrated under reduced pressure. The residue was purified by flash column chromatography using petroleum ether and ethyl acetate as eluent to afford coupling products.

## Data availability
Data available in this study are provided in the supplementary information. Crystallographic data coordinates for structures reported in this article has been deposited at the Cambridge Crystallographic Data Center (CCDC), under deposition numbers CCDC 2224253 (Mn complex 174). The sedata can be obtained free of charge from the Cambridge Crystallographic Data Center via https://www.ccdc.cam.ac.uk/structures/.

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

## Acknowledgements

This research is supported by the National Natural Science Foundation of China (Grant No. 22171174, 22471150 to D.X.; Grant No. 22402113 to G.S.), the Fundamental Research Funds for the Central Universities (Grant No. GK202406026 to T.K.; Grant No. GK202505023 to G.S.; Grant No. GK202505026 to D.X.), the Innovation Capability Support Program of Shaanxi (Grant No. 2023-CX-TD-28 to D.X.), the Fundamental Science Research Project of Shaanxi for Chemistry, Biology (Grant No. 22JHZ0027 to D.X.), the Natural Science Foundation of Shaanxi Province (Grant No. 2024JC-YBQN-0075 to G.S.; Grant No. 2025JC-YBQN-144 to T.K.), the State Key Laboratory of Natural and Biomimetic Drugs (Grant No. K202437 to G.S.), the Key Research and Development Program of Shaanxi (Grant No. 2024CY-JJQ-26 to D.X.), and the S&T Program of Energy Shaanxi Laboratory (Grant No. ESLB202420 00 to D.X.).

## Author contributions

D.X. conceived and directed the project. G.S. designed and conducted experiments, analyzed data. J.S. participated in the substrate scope expansion. Q.L., X.S., X.L., D.P., J.F., and H.S. helped analyzed data. T.K., J.D., G.L., and C.W. contributed to the project discussion. G.S. prepared the manuscript. D.X. wrote the manuscript. All authors discussed the experimental results and commented on the manuscript.

## Competing interests

The authors declare no competing interests.
