## [Transparent Peer Review file · Nature Communications]

Photoinduced Mn Catalysis for Efficient Platform for C-Heteroatom Bond Coupling of Aryl Halides

Corresponding Author: Professor Dong Xue

Version 0:

Reviewer comments:

Reviewer #1

(Remarks to the Author)

Photoinduced transition-metal catalysis provides innovative strategies for promoting novel chemical reactions and improving existing ones. In this manuscript, Xue and collaborators present an efficient and practical approach for achieving C–heteroatom bond coupling of aryl halides via photoinduced manganese catalysis. The work expands the synthetic toolbox for C(sp²)–N, C(sp²)–O, and C(sp²)–S bond formations. It is particularly noteworthy that, a single Mn(II)–bipyridine complex functions simultaneously as both a light-harvester and metal catalyst under light induction, providing flexibility in reaction design. Furthermore, the method is exceptionally well-developed for manganese-catalyzed C–heteroatom coupling reactions, with a broad substrate scope (>150 examples), excellent functional group tolerance, and compelling applications in late-stage functionalization of bioactive compounds. Furthermore, the method accommodates eight different nitrogen sources for C–N coupling, as well as C–O coupling with alcohols and C–S coupling with thiophenols, achieving yields of up to 94%. Its application in the late-stage functionalization of bioactive compounds and natural products offers a valuable new approach for the synthesis of these important molecules in medicinal chemistry and related fields. In mechanistic studies, the existence of Mn(I) species was indirectly confirmed by electron paramagnetic resonance experiments and control experiments, providing reasonable support for Mn(I)-mediated catalytic coupling reactions. The manuscript and supporting information are generally well-organized and clearly presented. Given the high significance and overall quality of this contribution, I believe that this work indeed possesses the expected level of novelty sufficient to justify its publication in Nature Communications. I recommend its acceptance by Nature Communications. Before formal acceptance, I have a few points of curiosity and minor suggestions:

1. Can aryl iodides be substituted in place of aryl bromides (chlorides)? It may be useful when the corresponding aryl bromides (chlorides) are not commercially available. If it fails, it does not affect the importance of this work.
2. Although in this manuscript demonstrate good substrate compatibility, I still recommend that the authors present some unsuccessful substrate examples in SI to provide readers with a more comprehensive understanding.
3. In C–O coupling reactions, can phenol or water serve as coupling partners?
4. Heterocycles are extremely important in drug molecules. In this manuscript, heteroaryl halides containing furan or thiophene were not found. How do their reactions perform?
5. The author previously developed a series of nickel-catalyzed reactions; compared with those systems, how are the advantages of manganese catalysis manifested?
6. I noticed several grammatical errors, which I won't enumerate here, but I recommend the authors conduct a thorough and meticulous proofreading of the entire text.
7. In the substrate scope section, the bond colors for the compounds are inconsistent. Please check and revise.
8. In the mechanistic cycle, the nucleophile should not be represented as NH₂R; it should be changed to NuH, because nucleophiles include alcohols, thiophenols, etc., and are not limited to amine compounds.

Reviewer #2

(Remarks to the Author)

Xue and coworkers demonstrated a manganese-based photocatalytic system for general C-heteroatom bond formation. The protocol is effective for couplings between aryl halides—encompassing even the challenging aryl chlorides—and a wide range of nucleophiles, demonstrating considerable substrate generality. Notably, this work unveils a new reactivity of manganese catalysts in this context, despite the prior validation of analogous Ni and Co catalysts by the same research group. The authors conducted mechanistic investigations, proposing a MnI/MnIII manifold as responsible for the catalytic

activity. I think the work would be of interest to the communities focused on metal catalysis and organic synthesis. However, several issues should be addressed before the manuscript can be considered for acceptance.

1# The substrate scope demonstrates good reactivity with alkyl primary amines, leading to the formation of N-alkyl arylamines, and no over-arylation products were reported. It is recommended that the authors perform DFT calculations to rationalize the lower reactivity of secondary arylamines.

2# To better showcase the applicability of this method, the use of more complex amines should be considered, rather than limiting the examples to modifications on the aryl halide moiety, as seen in Scheme 3 (e.g., compounds 157, 158, 159).

3# The manuscript highlights the excellent reactivity of the manganese catalyst in C–heteroatom coupling reactions. While this is commendable, such coupling reactions have already been achieved with several other metals. The current methodology does not appear to offer a distinct advantage over previously reported methods (for instance, cobalt catalysis from the same research group). Therefore, the authors are encouraged to incorporate additional elements that emphasize the unique benefits or novelty of manganese catalysis, if possible.

4# The authors consistently employed a reaction temperature of 85°C. It would be valuable to clarify which step—oxidative addition or reductive elimination—requires such high energy input. Identifying the rate-determining step would further strengthen the mechanistic discussion.

5# The proposed mechanism involves a Mn^I/Mn^{III} catalytic cycle for the coupling reaction. How would the system perform with ArOTf substrates, which typically exhibit reactivity similar to ArBr? Such results are not included in the manuscript. Additionally, it would be insightful to determine whether the oxidative addition proceeds via a single-electron or two-electron process.

Reviewer #3

(Remarks to the Author)

Abundant-metal-catalyzed cross-coupling reactions of aryl halides with nucleophiles to form C-heteroatom bonds are a core technology in modern synthesis, significantly accelerating the process of molecular diversification. Manganese, as one of the earth-abundant transition metals in the Earth's crust, offers a promising strategy for catalyzing C–heteroatom bond formations, serving as a potential alternative to the more established precious metal–catalyzed processes in organic synthesis. However, many of the manganese-catalyzed C–X bond coupling reactions are in fact triggered by copper impurities, causing progress in Mn-catalyzed C–heteroatom bond coupling reactions to largely stagnate. In this manuscript, Xue and co-workers present an excellent work on light-induced manganese catalysis for C-heteroatom bond coupling of aryl halides. This work has done ICP-MS analysis and further control experiments to exclude other metals' influence. The authors discovered a single Mn(II)–bipyridine complex functions simultaneously as both a light-harvester and metal catalyst under light induction is particularly noteworthy, providing flexibility in reaction design. This method exhibits a broad substrate scope of over 150 examples, enabling the coupling of diverse aryl halides with various nucleophiles to afford structurally diverse products, and demonstrates high functional group tolerance. Furthermore, it shows good compatibility with late-stage functionalization and modification of bioactive molecules and natural products, providing a valuable new synthetic approach for these important compounds in medicinal chemistry and related fields. In addition, the authors investigated the detailed reaction mechanism. Control experiments demonstrated that the Mn(I) species serves as the active species responsible for maintaining and initiating the reaction, and that the coupling product is formed through reductive elimination involving a possible Mn(III) intermediate species. It is proposed that the reaction proceeds via an Mn(I)/Mn(III) catalytic cycle. The synthetic applicability and robustness of this reaction have been clearly demonstrated, providing new insights into the development of Mn(I) chemistry and potentially establishing it as one of the landmark works in the field of manganese catalysis. Considering all the above, I recommend this work for publication in Nature Communications. Several minor revisions should be made before the formal acceptance.

1. Given that aryl halides have a broad substrate scope, I am curious about the reaction outcomes for sterically hindered aryl halides. Specifically, what are the results for substrates such as 2-tert-butyl or 2-isopropyl aryl halides (bromides, iodides, chlorides) and 2,6-disubstituted aryl halides (bromides, iodides, chlorides)?

2. During the evaluation of amine nucleophilic, the authors studied various aliphatic amines; Could simple tert-butylamine be used as nucleophiles?

3. A very recent reviews on Mn catalysis (CCS Chem. 2024, 6, 537–584) might be involved in the references.

4. In scheme 1B, Mn-catalyzed C-N cross-coupling reactions should be changed to Mn-catalyzed C-N cross-coupling reactions under thermal conditions.

5. In Table S8. the analysis may be performed on ICP-MS, not IPC-MS.

6. SI: the authors show a comparison of Mn salts from different vendors in Table S13, but then at the general procedure they mention a different vendor.

Version 1:

Reviewer comments:

Reviewer #1

(Remarks to the Author)

The authors have successfully addressed all the issues raised. I recommend the manuscript for acceptance and publication in Nature Communications.

Reviewer #2

(Remarks to the Author)

Concerns have been adequately addressed.

Reviewer #3

(Remarks to the Author)

The author has adequately addressed my concern. I recommend publication as is.

Responses to the Comments by Reviewer: 1

General Comment:

Photoinduced transition-metal catalysis provides innovative strategies for promoting novel chemical reactions and improving existing ones. In this manuscript, Xue and collaborators present an efficient and practical approach for achieving C–heteroatom bond coupling of aryl halides via photoinduced manganese catalysis. The work expands the synthetic toolbox for C(sp²)–N, C(sp²)–O, and C(sp²)–S bond formations. It is particularly noteworthy that, a single Mn(II)–bipyridine complex functions simultaneously as both a light-harvester and metal catalyst under light induction, providing flexibility in reaction design. Furthermore, the method is exceptionally well-developed for manganese-catalyzed C–heteroatom coupling reactions, with a broad substrate scope (>150 examples), excellent functional group tolerance, and compelling applications in late-stage functionalization of bioactive compounds. Furthermore, the method accommodates eight different nitrogen sources for C–N coupling, as well as C–O coupling with alcohols and C–S coupling with thiophenols, achieving yields of up to 94%. Its application in the late-stage functionalization of bioactive compounds and natural products offers a valuable new approach for the synthesis of these important molecules in medicinal chemistry and related fields. In mechanistic studies, the existence of Mn(I) species was indirectly confirmed by electron paramagnetic resonance experiments and control experiments, providing reasonable support for Mn(I)-mediated catalytic coupling reactions. The manuscript and supporting information are generally well-organized and clearly presented. Given the high significance and overall quality of this contribution, I believe that this work indeed possesses the expected level of novelty sufficient to justify its publication in *Nature Communications*. I recommend its acceptance by *Nature Communications*. Before formal acceptance, I have a few points of curiosity and minor suggestions:

Response: We thank the reviewer's valuable advice. The manuscript has been carefully revised based on your comments and suggestions.

Specific Comment

1. Can aryl iodides be substituted in place of aryl bromides (chlorides)? It may be useful when the corresponding aryl bromides (chlorides) are not commercially available. If it fails, it does not affect the importance of this work.

Response: We thank the reviewer's valuable advice. According to the reviewer's suggestion, we have tried aryl iodide as aryl electrophilic in the C-N coupling reaction. Notably, aryl iodide can efficiently undergo coupling reactions (see below), and the relevant discussion has been added to the manuscript.

^aReaction conditions: Aryl iodides (0.2 mmol), amine (0.4 mmol), $\text{Mn}(\text{OAc})_2$ (5.0 mol%), *d*-Mebpy (5.0 mol %), DBU (1.5 equiv., 0.3 mmol), DMAc (2.0 mL), purple LEDs (390–395 nm), 85 °C, under Ar, 24 h. Isolated yield.

2. Although in this manuscript demonstrate good substrate compatibility, I still recommend that the authors present some unsuccessful substrate examples in SI to provide readers with a more comprehensive understanding.

Response: We thank the reviewer's valuable advice. We have supplemented unsuccessful examples in the SI.

3. In C–O coupling reactions, can phenol or water serve as coupling partners?

Response: We thank the reviewer's valuable advice. According to the reviewer's suggestion, we attempted to conduct the reaction using phenol or water as the nucleophile. However, the yield of the C–O coupling product with phenol was relatively low, and water afforded almost no coupling product, and related studies are currently underway.

4. Heterocycles are extremely important in drug molecules. In this manuscript, heteroaryl halides containing furan or thiophene were not found. How do their reactions perform?

Response: We thank the reviewer's valuable advice. According to the reviewer's suggestion, we have attempted the use of furan or thiophene as heteroaryl halides in the reaction. However, almost no corresponding coupling products were obtained.

5. The author previously developed a series of nickel-catalyzed reactions; compared with those systems, how are the advantages of manganese catalysis manifested?

We thank the reviewer's valuable advice. As elaborated in our manuscript, manganese-based catalysts possess several distinct advantages over other transition metal catalysts, which are summarized as follows: 1) Earth abundance and cost-effectiveness: Manganese ranks as the third most abundant transition metal in the Earth's crust, rendering it considerably more accessible and cost-effective than precious metals (e.g., palladium, platinum, rhodium) that are widely employed in cross-coupling reactions. 2) Low toxicity and environmental compatibility: Compared to numerous other transition metal catalysts, manganese exhibits excellent biocompatibility and low toxicity. It is compatible with pharmaceutical applications where metal contamination constitutes a critical concern, and it possesses a lower bioaccumulation potential relative to heavy metals. 3) Broad substrate scope and high functional group tolerance: This synthetic method features an extensive substrate scope encompassing over 150 examples, enabling the coupling of diverse aryl halides with various nucleophiles to

afford structurally diverse products. Additionally, it demonstrates remarkable tolerance towards a wide range of functional groups. 4) Mechanistic insights into Mn(I) chemistry: Experimental studies have confirmed that Mn(I) species act as the active catalytic species responsible for initiating and sustaining the reaction cycle. These findings provide novel mechanistic insights for the advancement of Mn(I)-mediated catalytic systems and hold substantial potential for establishing new synthetic paradigms in cross-coupling chemistry.

Owing to these distinctive merits, manganese-based catalysts represent a highly promising platform for promoting sustainable development in the field of cross-coupling reactions. Corresponding discussions have been supplemented in the revised manuscript.

6. I noticed several grammatical errors, which I won't enumerate here, but I recommend the authors conduct a thorough and meticulous proofreading of the entire text.

Response: We thank the reviewer's valuable advice. We have revised the error in the manuscript.

7. In the substrate scope section, the bond colors for the compounds are inconsistent. Please check and revise.

Response: We thank the reviewer's valuable advice. We have revised the error in the manuscript.

8. In the mechanistic cycle, the nucleophile should not be represented as NH_2R ; it should be changed to NuH , because nucleophiles include alcohols, thiophenols, etc., and are not limited to amine compounds.

Response: We thank the reviewer's valuable advice. **Response:** We thank the reviewer's valuable advice. We have revised the nucleophile changed to NuH in the reaction mechanism cycle in the manuscript.

Responses to the Comments by Reviewer: 2

General Comment:

Xue and coworkers demonstrated a manganese-based photocatalytic system for general C-heteroatom bond formation. The protocol is effective for couplings between aryl halides—encompassing even the challenging aryl chlorides—and a wide range of nucleophiles, demonstrating considerable substrate generality. Notably, this work unveils a new reactivity of manganese catalysts in this context, despite the prior validation of analogous Ni and Co catalysts by the same research group. The authors conducted mechanistic investigations, proposing a MnI/MnIII manifold as responsible for the catalytic activity. I think the work would be of interest to the communities focused on metal catalysis and organic synthesis. However, several issues should be addressed before the manuscript can be considered for acceptance.

Response: We thank the reviewer's valuable advice. The manuscript has been carefully revised based on your comments and suggestions.

Specific Comment

1. The substrate scope demonstrates good reactivity with alkyl primary amines, leading to the formation of *N*-alkyl arylamines, and no over-arylation products were reported. It is recommended that the authors perform DFT calculations to rationalize the lower reactivity of secondary arylamines.

Response: We thank the reviewer's valuable advice. Density functional theory (DFT) calculations were performed for the amine coordination process to the Mn(III) center, employing the computational level of SMD(DMAc)-PBE0-D3BJ/SDD&6-311+G(2d,p)//PBE0-D3BJ/SDD&6-31G(d). The calculation results indicate that the coordination of primary alkylamines to the Mn(III) center is endothermic with an energy change of 4.2 kcal/mol, whereas the coordination of secondary arylamines exhibits a significantly higher endothermic value of 15.2 kcal/mol. This distinct energy difference confirms that the coordination process of secondary arylamines is thermodynamically less favorable, which further impedes the subsequent reaction steps and consequently leads to their lower reactivity. Additionally, it should be noted that the comprehensive catalytic cycle at the experimental level is still under further

investigation, and relevant updates will be supplemented in future revisions as appropriate.

2. To better showcase the applicability of this method, the use of more complex amines should be considered, rather than limiting the examples to modifications on the aryl halide moiety, as seen in Scheme 3 (e.g., compounds 157, 158, 159).

Response: We thank the reviewer's valuable advice. According to the reviewer's suggestion, we have expanded the more complex amines as nucleophiles in Scheme 3 (Amino acid ester, dehydroabietylamine; see 172-173). These additional examples afford the corresponding *N*-alkyl arylamines in moderate to good yields. These new results further support the generality of our protocol beyond simple primary alkyl amines and better illustrate the synthetic utility of this transformation.

3. The manuscript highlights the excellent reactivity of the manganese catalyst in C-heteroatom coupling reactions. While this is commendable, such coupling reactions have already been achieved with several other metals. The current methodology does not appear to offer a distinct advantage over previously reported methods (for instance, cobalt catalysis from the same research group). Therefore, the authors are encouraged

to incorporate additional elements that emphasize the unique benefits or novelty of manganese catalysis, if possible.

Response: As elaborated in our manuscript, manganese-based catalysts possess several distinct advantages over other transition metal catalysts, which are summarized as follows: 1) Earth abundance and cost-effectiveness: Manganese ranks as the third most abundant transition metal in the Earth's crust, rendering it considerably more accessible and cost-effective than precious metals (e.g., palladium, platinum, rhodium) that are widely employed in cross-coupling reactions. 2) Low toxicity and environmental compatibility: Compared to numerous other transition metal catalysts, manganese exhibits excellent biocompatibility and low toxicity. It is compatible with pharmaceutical applications where metal contamination constitutes a critical concern, and it possesses a lower bioaccumulation potential relative to heavy metals. 3) Broad substrate scope and high functional group tolerance: This synthetic method features an extensive substrate scope encompassing over 150 examples, enabling the coupling of diverse aryl halides with various nucleophiles to afford structurally diverse products. Additionally, it demonstrates remarkable tolerance towards a wide range of functional groups. 4) Mechanistic insights into Mn(I) chemistry: Experimental studies have confirmed that Mn(I) species act as the active catalytic species responsible for initiating and sustaining the reaction cycle. These findings provide novel mechanistic insights for the advancement of Mn(I)-mediated catalytic systems and hold substantial potential for establishing new synthetic paradigms in cross-coupling chemistry.

Owing to these distinctive merits, manganese-based catalysts represent a highly promising platform for promoting sustainable development in the field of cross-coupling reactions. Corresponding discussions have been supplemented in the revised manuscript.

4. The authors consistently employed a reaction temperature of 85°C. It would be valuable to clarify which step—oxidative addition or reductive elimination—requires such high energy input. Identifying the rate-determining step would further strengthen the mechanistic discussion.

Response: We thank the reviewer's valuable advice. The relatively high reaction temperature (85 °C) is mainly required to overcome the barrier associated with the oxidative addition step. In this step, the metal center must insert into the C–X of the substrate, which constitutes the bond activation process. Such oxidative addition typically involves cleavage of a strong covalent bond and formation of a new metal–carbon bond, both of which demand a considerable amount of energy to occur efficiently. In our previous work we also found that low-valent metals require a certain amount of heat to overcome the activation barrier for oxidative addition (*Angew. Chem. Int. Ed.* **2021**, *61*, 21536; *J. Org. Chem.* **2022**, *87*, 10285; *Angew. Chem. Int. Ed.* **2024**, *63*, e202314355; *J. Am. Chem. Soc.* **2024**, *146*, 26936; *Nat. Commun.* **2025**, *16*, 7045). The experimental observations also support this explanation: the reaction proceeds slowly at lower temperatures and no oxidation addition product **162**. However, when the temperature is raised to 85–90 °C, the **162** can be obtained in moderate yield, indicating that the Mn(I) species requires a certain temperature to overcome the activation barrier for oxidative addition. Once the oxidative addition intermediate is formed, the subsequent reductive elimination can proceed smoothly without additional heating (**Scheme 4. D**). To further demonstrate that oxidative addition is the rate-determining step of this transformation, we conducted a Hammett analysis on a series of para-substituted aryl substrates. A clear linear free-energy relationship was observed between the substituent constants and the measured rate data. Electron-withdrawing substituents were found to accelerate the reaction, whereas electron-donating groups slowed it down. This trend indicates a build-up of positive character at the reaction center in the transition state, which is consistent with an oxidative addition process involving bond activation at the metal center. These results support our mechanistic proposal that oxidative addition is the rate-determining step, while the subsequent reductive elimination proceeds with a significantly lower energy barrier.

5. The proposed mechanism involves a Mn^I/Mn^{III} catalytic cycle for the coupling reaction. How would the system perform with ArOTf substrates, which typically exhibit reactivity similar to ArBr? Such results are not included in the manuscript. Additionally, it would be insightful to determine whether the oxidative addition proceeds via a single-electron or two-electron process.

Response: We thank the reviewer's valuable advice. According to the reviewer's suggestion, we have tried ArOTf substrates as aryl electrophilic in the C-N coupling reaction. Most ArOTf derivatives afforded the desired coupling products in moderate to good yields (see below), comparable to those obtained from the corresponding aryl bromides. Those results indicate that the present Mn-catalyzed system exhibits broad electrophile compatibility, tolerating both halides and pseudohalides. Based on the experimental results for ArOTf derivatives, it was further confirmed that the oxidative

addition step in this method proceeds predominantly via a two-electron process. And the corresponding data and discussion have now been added to in the manuscript.

^aReaction conditions: Aryl OTf (0.2 mmol), amine (0.4 mmol), Mn(OAc)₂ (15.0 mol%), *d*-Mebpy (15.0 mol %), DBU (1.5 equiv., 0.3 mmol), DMAc (2.0 mL), purple LEDs (390–395 nm), 85 °C, under Ar, 36.^b 48 h. Isolated yield.

Responses to the Comments by Reviewer: 3

General Comment:

Abundant-metal-catalyzed cross-coupling reactions of aryl halides with nucleophiles to form C-heteroatom bonds are a core technology in modern synthesis, significantly accelerating the process of molecular diversification. Manganese, as one of the earth-abundant transition metals in the Earth's crust, offers a promising strategy for catalyzing C-heteroatom bond formations, serving as a potential alternative to the more established precious metal-catalyzed processes in organic synthesis. However, many of the manganese-catalyzed C-X bond coupling reactions are in fact triggered by copper impurities, causing progress in Mn-catalyzed C-heteroatom bond coupling reactions to largely stagnate. In this manuscript, Xue and co-workers present an excellent work on light-induced manganese catalysis for C-heteroatom bond coupling of aryl halides. This work has done ICP-MS analysis and further control experiments to exclude other metals' influence. The authors discovered a single Mn(II)-bipyridine complex functions simultaneously as both a light-harvester and metal catalyst under light induction is particularly noteworthy, providing flexibility in reaction design. This method exhibits a broad substrate scope of over 150 examples, enabling the coupling

of diverse aryl halides with various nucleophiles to afford structurally diverse products, and demonstrates high functional group tolerance. Furthermore, it shows good compatibility with late-stage functionalization and modification of bioactive molecules and natural products, providing a valuable new synthetic approach for these important compounds in medicinal chemistry and related fields. In addition, the authors investigated the detailed reaction mechanism. Control experiments demonstrated that the Mn(I) species serves as the active species responsible for maintaining and initiating the reaction, and that the coupling product is formed through reductive elimination involving a possible Mn(III) intermediate species. It is proposed that the reaction proceeds via an Mn(I)/Mn(III) catalytic cycle. The synthetic applicability and robustness of this reaction have been clearly demonstrated, providing new insights into the development of Mn(I) chemistry and potentially establishing it as one of the landmark works in the field of manganese catalysis. Considering all the above, I recommend this work for publication in Nature Communications. Several minor revisions should be made before the formal acceptance.

Specific Comment

1. Given that aryl halides have a broad substrate scope, I am curious about the reaction outcomes for sterically hindered aryl halides. Specifically, what are the results for substrates such as 2-tert-butyl or 2-isopropyl aryl halides (bromides, iodides, chlorides) and 2,6-disubstituted aryl halides (bromides, iodides, chlorides)?

Response: We thank the reviewer's valuable advice. According to the reviewer's suggestion, we investigated the use of 2-tert-butyl, 2-isopropyl, 2,6-dimethoxy aryl halides and 2,4,6-trimethyl aryl halides as the aryl electrophiles for the C–N coupling reaction. Fortunately, the 2-isopropyl aryl iodides were able to undergo the coupling reaction with high efficiency (see below). Although the corresponding bearing 2-isopropyl aryl bromides also participated in the coupling, their reactivity was relatively lower. For aryl chlorides, likely due to their very low reactivity, essentially no desired product was obtained. In the case of 2,6-dimethoxy and 2,4,6-trimethyl aryl halides, they exhibited virtually no reactivity, which may be attributed to steric hindrance. And

the corresponding data and discussion have now been added to in the manuscript.

^aReaction conditions: Aryl halides (0.2 mmol), amine (0.4 mmol), Mn(OAc)₂ (15.0 mol%), *d*-Mebpy (15.0 mol %), DBU (1.5 equiv., 0.3 mmol), DMAc (2.0 mL), purple LEDs (390–395 nm), 85 °C, under Ar, 24 h. Isolated yield.

2. During the evaluation of amine nucleophilic, the authors studied various aliphatic amines; Could simple tert-butylamine be used as nucleophiles?

Response: We thank the reviewer's valuable advice. We have conducted tested tert-butylamine as a representative hindered aliphatic amine to evaluate the effect of steric hindrance on the reaction outcome. In this case, the coupling product was obtained in trace amount.

3. A very recent reviews on Mn catalysis (CCS Chem. 2024, 6, 537–584) might be involved in the references.

Response: We thank the reviewer's valuable advice. The reference has been added in the manuscript.

4. In scheme1B, Mn-catalyzed C-N cross-coupling reactions should be changed to Mn-catalyzed C-N cross-coupling reactions under thermal conditions.

Response: We thank the reviewer's valuable advices. We have revised the related statements in the manuscript.

5. In Table S8. the analysis may be performed on ICP-MS, not IPC-MS.

Response: We thank the reviewer's valuable advice. We have revised the error in the SI.

6. SI: the authors show a comparison of Mn salts from different vendors in Table S13, but then at the general procedure they mention a different vendor.

Response: We thank the reviewer's valuable advice. We present a comparison of Mn salts from different suppliers in Table S13, mainly due to concerns about possible trace metal effects, but we also mention vendor of catalysts used in the general procedure.

We believe the revision has addressed most, if not all, of the reviewers' concerns, and we look forward to hearing from you.

Thank you for your consideration and best regards,

Yours sincerely,

Dong Xue